

# Occurrence, abundance, and formation of atmospheric tarballs from a wide range of wildfires in the western US

Kouji Adachi[1], Jack E. Dibb[2], Joseph M. Katich[3,4#], Joshua P. Schwarz[3], Hongyu Guo[4,5], Pedro Campuzano-Jost[4,5], Jose L. Jimenez[4,5], Jeff Peischl[3], Christopher D. Holmes[6], James Crawford[7]

[1]Department of Atmosphere, Ocean, and Earth System Modelling Research, Meteorological Research Institute, Tsukuba, Japan
[2]Institute for the Study of Earth, Oceans, and Space, University of New Hampshire, Durham, NH, USA
[3]Chemical Sciences Laboratory, National Oceanic and Atmospheric Administration, Boulder, CO, USA
[4]Cooperative Institute for Research in Environmental Sciences, University of Colorado Boulder, Boulder, CO, USA
[5]Department of Chemistry, University of Colorado Boulder, Boulder, CO, USA
[6]Earth, Ocean, and Atmospheric Science, Florida State University, Tallahassee, FL, USA
[7]NASA Langley Research Center, Hampton, VA, USA
[#]now at Ball Aerospace, Boulder, CO, USA

*Correspondence to*: Kouji Adachi (adachik@mri-jma.go.jp)

**Abstract.** Biomass burning emits large numbers of organic aerosol particles, a subset of which are called tarballs (TBs). TBs possess spherical morphology and unique physical, chemical, and optical properties. They are recognized as brown carbon aerosol particles, thereby having implications for climate through the absorption of solar radiation. Aerosol particles were collected from wildfire and agricultural fire smoke sampled by the NASA DC-8 aircraft during the FIREX-AQ campaign in the western US from July to September 2019. The current study developed an image analysis method applying deep learning to distinguish TBs from other round particles that deformed on the substrate, based on their morphological characteristics in the transmission electron microscopy images. This study detected 4567 TBs with mostly <10 h downwind from the emissions and measured their compositions, abundance, sizes, and mixing states. The number fraction, mass fraction, and concentration of TBs from all wildfire smoke were 10% ± 1%, 10% ± 2%, and 10.1 ± 4.6 µg m$^{-3}$, respectively. As the samples aged from emission up to 5 h, the TB number fractions roughly increased from 5% to 15%, indicating that TBs are processed primary particles. In more aged samples, the fraction decreased possibly due to dilution and removal. We also showed TBs within pyrocumulonimbus (PyroCb) activity and various TB mixing states. This study reveals the abundances and physical and chemical properties of a wide range of TBs from various biomass-burning events and enhances the knowledge of TB emissions, which contributes to the evaluation of the climate impact of TBs.



## 1 Introduction

Open biomass burning (e.g., wildfire, agricultural fire) emits significant amounts of gases and particulate matter into the atmosphere (Yokelson et al., 2007; Andreae, 2019; Gkatzelis et al., 2024) and largely influences the global climate (Szopa et al., 2021) and human health (Karanasiou et al., 2021). While the future impacts of biomass burning are highly uncertain, a warmer climate will likely increase fire frequency and severity (Keywood et al., 2011; Abatzoglou and Williams, 2016; Jaffe et al., 2020), implying that they will significantly influence the future climate.

The influences of biomass burning on the climate include both cooling and warming effects. The emissions of light-scattering aerosol particles generally have a negative influence on the radiative forcing. In contrast, the emissions of greenhouse gases and light-absorbing aerosol particles enhance the positive radiative forcing (Szopa et al., 2021). As a result of both negative and positive effects, the overall climate effect of biomass burning is subject to a large degree of uncertainty (Szopa et al., 2021).

Brown carbon is a type of organic aerosol particle commonly emitted from biomass burning. It is characterized by an absorption spectrum that smoothly increases from visible to ultraviolet wavelengths (Laskin et al., 2015). Because of their light-absorbing properties, they have positive radiative forcing effects (Andreae and Gelencsér, 2006; Laskin et al., 2015). Brown carbon aerosol particles from biomass burning have various compositions, volatilities, optical properties, and formation processes, and this study focuses on a specific type of brown carbon aerosol particles called tarball (TB).

TBs were first recognized and defined in companion papers by Li et al. (2003) and Pósfai et al. (2003) from biomass burning aerosol samples in southern Africa using transmission electron microscopy (TEM). Pioneering studies have revealed their detailed chemical and physical properties (Pósfai et al., 2004; Hand et al., 2005). Since then, TBs have been reported in various locations across the globe, including biomass burning smoke (e.g., Adachi and Buseck, 2011; Adachi et al., 2019), urban areas (e.g., Fu et al., 2012; Sparks and Wagner, 2021), remote mountains (e.g., Yuan et al., 2020), the oceanic atmosphere

(e.g., Yoshizue et al., 2020), and the Arctic region (e.g., Moroni et al., 2017; Moroni et al., 2020; Adachi et al., 2021).

TBs are commonly defined based on their unique physical properties, such as spherical shapes and insensitivity to electron beams and heat (Pósfai et al., 2004; Hand et al., 2005; Chakrabarty et al., 2010; China et al., 2013; Adachi et al., 2017; Corbin and Gysel-Beer, 2019). Further TB measurements have also revealed that they have light-absorbing optical properties (Hand et al., 2005; Chakrabarty et al., 2010; Sedlacek, et al., 2018). Thus, it is hypothesized that they have notable climate

influences because of their light-absorbing properties and abundance of biomass burning emissions (Jacobson, 2012; Chakrabarty et al., 2023). In addition, studies suggest that TBs possibly act as ice-nucleating particles (INPs) (Barry et al., 2021) and adversely affect human health (Pardo et al., 2020). Despite their potential influences on various environmental aspects, their atmospheric abundance, formation processes, and influences on climate and human health are not fully understood and remain uncertain. A reason for this uncertainty is the difficulty of TB detection. Corbin and Gysel-Beer (2019)

demonstrated that TBs can be detected using an online measurement (single particle soot photometer). However, most studies manually identify TBs by observing their spherical shapes one by one in samples collected on a filter substrate using



microscopic techniques such as electron microscopy and synchrotron-based X-ray microspectroscopy (Tivanski et al., 2007; China et al., 2013).

The optical properties of TBs are of great interest in evaluating their climate influences. Thus, studies have attempted to evaluate their light-absorbing properties (Hand et al., 2005; Alexander et al., 2008; Chakrabarty et al., 2010; Tóth et al., 2014; Hoffer et al., 2016; Sedlacek et al., 2018; Chakrabarty et al., 2023). These results suggest that TBs have a wide range of light-absorbing properties. Although the current study does not aim to evaluate their optical properties directly, Chakrabarty et al. (2023) have shown a range of TB refractive indices across wavelengths from 350 to 1200 nm in samples collected on the ground during the current study.

The formation process of TBs is controversial. Thermal, chemical, and physical processes, such as polymerization of organic matter, condensation, photochemical processes, water loss, heat shock, and temperature changes, have been proposed to contribute to the TB formation (Pósfai et al., 2004; Laskin et al., 2015; Reid et al., 2018; Corbin and Gysel-Beer, 2019; Adachi et al., 2019). Laboratory experiments have also shown that TBs are produced by the direct emission of liquid tar droplets followed by heat transformation (Tóth et al., 2014). An aircraft-based sampling of biomass burning smoke and TEM measurements observed gradual processes of forming TBs or an organic viscosity increment within biomass burning smoke (Pósfai et al., 2004; Adachi and Buseck, 2011; Sedlacek et al., 2018; Adachi et al., 2019). A previous study (Adachi et al., 2019) proposed that chemical processes involving oxygen and nitrogen addition could contribute to TB formation or occur together with the TB formation. They suggested that more observations are necessary to support the hypothesized processes.

We collected biomass burning smoke samples during the NOAA/NASA Fire Influence on Regional to Global Environments and Air Quality (FIREX-AQ) campaign (Warneke et al., 2023). The FIREX-AQ campaign explored aerosol and gas emissions from wildfires and agricultural fires using aircraft, satellite remote sensing, modeling, and ground-based measurements over the western and southeastern US during the summer of 2019. Results from the FIREX-AQ campaign include observations of the physical, chemical, and optical properties of aerosol particles (Junghenn Noyes et al., 2020; Moore et al., 2021; Sumlin et al., 2021; Adachi et al., 2022a; Zeng et al., 2022; Chakrabarty et al., 2023; Katich et al., 2023; Pagonis et al., 2023; Siemens et al., 2024), ozone chemistry (Xu et al., 2021), evolution of volatile organic compounds (Decker et al., 2021; Liao et al., 2021), and emission factors (Travis et al., 2023) from biomass burning emissions.

Our previous study using the TEM samples from the FIREX-AQ campaign showed that ash-bearing particles characterized by Mg and Ca were abundant in fine particles (Adachi et al., 2022a). The current study used the same set of TEM samples and analyzed the data, focusing on TBs by developing an image analysis method to automatically detect TBs. The objectives of this study are to 1) develop a TB detection method from TEM images, 2) provide atmospheric observations of TBs collected from a wide range of wildfires and aging, and 3) discuss TB occurrences, including mixing states, abundances, mass concentrations, formation processes, and cloud interactions to evaluate their overall potential climate influences.





## 2 Methods

### 2.1 The FIREX-AQ campaign

The FIREX-AQ campaign was conducted from 24 July to 16 August based in Boise, Idaho, to measure wildfire and from 19 August to 3 September based in Salina, Kansas, to measure prescribed and agricultural fire (Table 1). This campaign used four aircraft, four mobile laboratories, and six ground observatories (Warneke et al., 2023). We used samples collected from the NASA DC8. The DC8 is an aircraft with various instruments and conducted a total of 20 flights during the campaign. We measured 221 TEM grids (53,727 particles) from nine flights that included relatively intense wildfire events and

agricultural fires (Adachi et al., 2022a).

**Table 1. TEM samples and detected TB information for wildfire and agricultural fire measured in this study.**

| Date | Main fires | State | Latitude | Longitude | TEM sample # | Particle # | TB # | TB number fractions (%) | Final burned area (acres)[a] | Primary fuels[a,b] |
|---|---|---|---|---|---|---|---|---|---|---|
| 25-Jul | Shady | ID | 44.52 | 115.02 | 22 | 6130 | 619 | 10 | 6,091 | Timber litter and shrubs under Douglas-fir, Pacific ponderosa-lodgepole pine/oceanspray forest |
| 06-Aug | Horsefly | MT | 46.96 | 112.44 | 24 | 6317 | 818 | 13 | 1,352 | Subalpine fir-lodgepole pine-whitebark pine-Engelmann spruce forest |
| 07-Aug | Williams Flats | WA | 47.94 | 118.62 | 24 | 5873 | 80 | 1 | 44,360 | Ignited in primarily Idaho fescue-bluebunch wheatgrass grassland and expanded to primarily Douglas-fir-Pacific ponderosa pine/oceanspray forest |
| 08-Aug | | | | | 21 | 5430 | 163 | 3 | | |
| 12-Aug | Castle | AZ | 36.53 | 112.23 | 24 | 6062 | 761 | 13 | 19,378 | Ponderosa pine-two-needle pinyon-Utah juniper forest |
| 13-Aug | | | | | 24 | 5660 | 1112 | 20 | | |
| 15-Aug | Sheridan | AZ | 34.68 | 112.89 | 22 | 5139 | 451 | 9 | 21,483 | Pinyon-Utah juniper forest |
| 16-Aug | | | | | 24 | 5801 | 505 | 9 | | |
| 03-Sep | Agricultural biomass burning[c] | IL, MO, AR, MS | 33.5-37.5 | 88.6-91.7 | 36 | 7315 | 58 | 1 | | Crop residue (rice and corn) |
| Total | | | | | 221 | 53727 | 4567 | 9 | | |

a: Warneke et al. (2023)

b: Fuel characteristic classification system (FCCS) name except the agricultural biomass burning (3-Sep)

c: Agricultural biomass burning samples were collected from various occurrences of smoke in the region.

### 2.2 Sample age estimate

The plume age during the sampling of the TEM samples was estimated from air parcel trajectories computed in the

HYSPLIT (Hybrid Single-Particle Lagrangian Integrated Trajectory) model (Stein et al., 2015) with multiple high-resolution meteorological datasets (HRRR 3 km, NAM CONUS Nest 3 km, and GFS 0.25°). Typical uncertainties in the plume age originate from, for example, the difference between observed and archived winds and are 25% of the estimated age (Decker et al., 2021). Other studies have also provided details of the sample age estimates during the FIREX-AQ campaign (Xu et al., 2021; Decker et al., 2021; Zeng et al., 2022; Warneke et al., 2023).





## 2.3 TEM sample collection


We collected smoke and background (non-smoke) aerosol particle samples using an impactor sampler (AS-24W, Arios Inc., Tokyo, Japan) (Adachi et al., 2022b; Adachi et al., 2023), which was equipped with two TEM grids containing overlapping Lacy carbon (top; U1001, EM-Japan, Tokyo, Japan) and Formvar substrates (bottom; U1007, EM-Japan, Tokyo, Japan). This study primarily used the Formvar substrates, with small and large 50% cutoff sizes of aerodynamic diameters of 100 and 700 nm, respectively. Sampling was performed to cover each transect of biomass burning smoke, with sampling times of ~1 to 3 min and an airflow rate of 1.0 L/min.


## 2.4 TEM analysis

A transmission electron microscope (JEM-1400, JEOL, Tokyo, Japan) equipped with an energy-dispersive X-ray spectrometer (EDS; X-Max 80, Oxford Instruments, Tokyo, Japan) was used in the TEM and scanning TEM (STEM-EDS) modes. The STEM-EDS measurements were performed using an acceleration voltage of 120 keV and an acquisition time of 20 s. Approximately 30 TEM images were initially taken across the sample grid to measure the particle shapes and select the representative areas. Then, the areas of approximately 100 representative particles were measured in the STEM mode at a magnification of ×6000. Appropriate thresholds were applied to dark-field STEM images to identify and distinguish aerosol particles from the substrate (Adachi et al., 2019). Characteristics of particles, such as area-equivalent diameter, shape factor, and EDS spectra, were determined from the measured particles. Area-equivalent diameters of TBs can be approximated to the volume-equivalent diameters as they retain spherical shapes. However, when the collected particles deform or spread on the substrate, the area-equivalent diameter tends to be approximately twice the volume-equivalent diameter (Zhang et al., 2020). These assumptions were applied to estimate the particle volumes. Volatile and semi-volatile particles may be lost during sampling and in the vacuum TEM chamber. The smallest particle cutoff size for STEM-EDS analysis was 0.25 μm in area-equivalent diameter (number of pixels > 100). The STEM-EDS provided relative weight percentages among the selected elements (C, N, O, Na, Mg, Al, Si, P, S, Cl, K, Ca, Ti, Mn, Fe, and Zn) within each particle. We estimated the possible N and O wt% within TBs and other organic (carbonaceous) particles originating only from their organic compounds. This calculation was done by assuming that all sulfur forms ammonium and potassium sulfates and that any leftover N and O other than sulfate were associated with organic matter (called "non-sulfate" N and O), i.e., we assumed that all Cl, K, S, N, and O in aerosol particles occur as follows: $[Cl] = KCl$; $[K] = KCl + K_2SO_4$; $[S] = K_2SO_4 + (NH_4)_2SO_4$; $[N] = (NH_4)_2SO_4 + [N \text{ in non-sulfate}]$; $[O] = [O \text{ in non-sulfate}] + [O \text{ in } K_2SO_4 + (NH_4)_2SO_4]$. Further discussion of this calculation and uncertainties have been provided by Adachi et al. (2019). The previous study used normalized O/K and N/K values to avoid the influence of uneven lacey carbon substrates. In the current study, we used the O and N values directly without normalizing them by K because we used flat formvar substrates, and the influence of C and O from the substrate did not significantly influence the measured wt% of other elements. Detection limits were typically 0.02 wt%, derived from one sigma of the measured peak intensities. Adachi et al. (2022a) also described details of the TEM measurements.



**2.5 Tarball detection using image analysis with deep learning and particle classification**

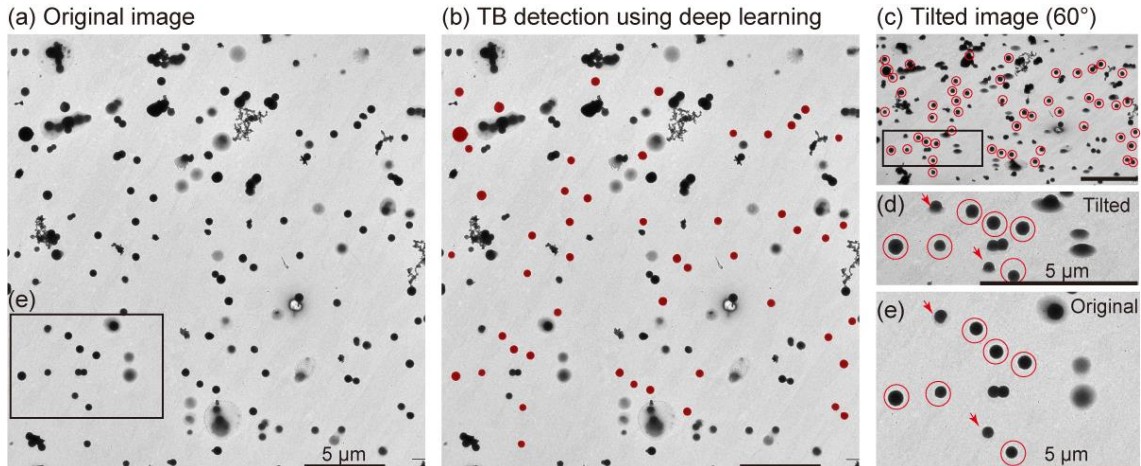

**Figure 1. TB detections from TEM images using the deep learning method. (a) The original TEM image. This sample**
**was collected from the Castle fire (14 August, 00:37 UTC). (b) TB detection using the deep learning method. Detected**
**TBs are shown in red. (c) The tilted image of the same area as the original image. The sample was tilted at 60° using a**
**tomography TEM holder. The detected TBs are marked using red circles. (d) The magnified tilted image of the selected**
**area in (c). (e) The magnified image of the selected area in (a). Images (d) and (e) show the same area. Other organic**
**particles spread over the substrates, whereas TBs retain their sphericity. Round particles deformed on the substrate**
**are excluded from the TBs in the deep learning method (red arrows in (d) and (e)). All scale bars indicate 5 µm.**

Tarballs are generally identified by their spherical shape in TEM images. However, when using a flat formvar
substrate, other low-viscosity organic particles can also appear round in the TEM image as they spread uniformly over the
substrates. Here, we use the contrast of TBs in TEM images differing from that of spreading organic particles, i.e., TB particles
are thicker and darker than the spreading organic particles due to their sphericity on the substrate (Fig.1). We used deep learning
image analysis software (MIPAR™ version 3.3.4) to identify TBs based on their morphological and imaging features (e.g.,
round and dark) in TEM images. The advantages of deep learning image analysis over the conventional technique are 1) it can
detect TB much quicker than the conventional technique (i.e., ~ seconds vs. minutes for a TEM image) and 2) it does not
depend on the experience of the operator, whereas the conventional method needs substantial training and experience to
unambiguously identify TBs.

First, 49 TEM images with relatively abundant TB particles from different samples during the FIREX-AQ campaign
were selected as teaching images. These TEM images contained 2534 TB particles that were manually identified using
conventional imaging techniques (i.e., extraction of spherical particles with high contrast (Adachi et al., 2019)). We constructed
a TB model in the software using these TB images and applied the model to the TEM images containing particles used for
STEM-EDS analyses (553 images). Then, we extracted particles larger than 0.15 µm, with roundness >0.95 and roughness <



1.05. Finally, the extracted particles were marked as TB in the corresponding particles measured in the STEM-EDS analysis. As a result, we obtained 4567 TB particles with compositional information. All detected TBs were manually checked using TEM images with marked TBs (Figs 1a and 1b) to confirm the image processes. Note that some TBs may be overlooked, especially the coagulated and slightly deformed ones (e.g., Figs. 1d and 1e) because our TB definition is relatively strict

compared with conventional imaging analysis. We used this definition to unambiguously select TBs to obtain their averaged compositions and sizes. Thus, the estimated TB number fraction is likely to be lower than that defined by conventional imaging analysis based solely on their spherical shape. In addition, we manually checked the presence of aggregated TBs only for the aggregation measurements in Section 3.1.

In addition to the image analysis used for TB identification, we classified all particles into six categories based on

their composition. The classification criteria are the same as those in Adachi et al. (2022a): 1) ash-bearing particles (Mg, Ca > 0.5 wt%), 2) mineral dust-bearing particles (Al, Fe > 0.5 wt%), 3) K-bearing particles (K > 2 wt%), 4) sulfate-bearing particles (S > 2 wt%), 5) carbonaceous particles without major inclusions (C + O > 90 wt%), and 6) others (none of the above). When particles were mixtures of two or more particle types (e.g., mineral dust mixed with sulfate), they were classified into a single category as shown in a flow chart (Fig. S1).

We classified all as having attributes of up to seven classes by combining the image analysis (TB identification, primary category) and compositional classification (secondary category): 1) TBs, 2) ash-bearing particles, 3) mineral dust-bearing particles, 4) K-bearing particles, 5) sulfate-bearing particles, 6) carbonaceous particles, and 7) others (Fig. S1). In this study, the term "carbonaceous particle" is chemically defined and excludes TBs, although TBs are carbonaceous particles. The term "organic particles" is structurally defined based on their amorphous nature in TEM images. The term "other organic

particles" refers to organic particles other than TBs.

## 2.6 Estimation of the tarball mass concentrations and enhancement ratios

In addition to the TB number fractions, atmospheric TB mass concentration ($TB_{mc}$, µg m$^{-3}$) and TB enhancement ratios relative to carbon monoxide ($TB_{mc}/dCO$) (i.e., $TB_{mc}$ divided by $dCO$ (ppb), which is $CO_{measured} - CO_{background}$) are useful for evaluating the TB climate effects (Yokelson et al., 2013). Although we did not directly measure atmospheric TB

concentrations, we evaluated these values based on our measurements (TB volume fraction) with assumptions such as aerosol densities and particle volumes. Although the estimated values include significant uncertainties originating from, for example, particle collection efficiency (e.g., bouncing effects (Bateman et al., 2017) and loss of volatile particles), the attempt will provide an idea of how much TB particles are emitted from biomass burning smoke and the changes that occur with aging.

First, individual aerosol particle volumes are determined based on particle sizes from TEM measurements. We

assumed that the TB area-equivalent diameters are the same as the volume-equivalent diameters. Conversely, we assumed that the area-equivalent diameters of non-TB particles became two times larger than their volume-equivalent diameters because non-TB particles were highly deformed on the substrates (Zhang et al., 2020). Second, the densities of TB, ash-bearing particles, dust-bearing particles, K-bearing particles, sulfate-bearing particles, carbonaceous particles, and other particles are assumed



to be 1.40 (Alexander et al., 2008), 2.70 (as calcium carbonate) (Adachi et al., 2022a), 2.70 (Salcedo et al., 2006), 2.66 (as
potassium sulfate) (Patnaik, 2003), 1.75 (Salcedo et al., 2006), 1.20 (Salcedo et al., 2006), and 2.00, respectively. We assume
that each particle comprises a single component. Finally, the TB mass fractions estimated from the TEM measurements were
applied to the atmospheric aerosol masses obtained by an aerosol mass spectrometer (AMS) to evaluate the $TB_{mc}$ by assuming
that the TB volume fractions estimated from our TEM measurements represent those in the sampled air. We used the dataset
of the Aerodyne high-resolution time-of-flight aerosol mass spectrometer (HR-ToF-AMS) shown in Adachi et al. (2022a). The
AMS had a 50% cutoff size of approximately 870 nm in aerodynamic diameter for typical FIREX-AQ plumes, larger than that
of the TEM sampler (~700 nm) (Adachi et al., 2022a). A comparison between TEM measurements and AMS results has been
discussed by Adachi et al. (2022a), showing reasonable agreement with uncertainties (e.g., ~50% in sulfate mass). CO mixing
ratios in dry air mole fraction were measured by an off-axis integrated cavity output spectrometer (Eilerman et al., 2016;
Warneke et al., 2023). Background CO concentrations were estimated from the CO concentrations of air without (or with
minimal) biomass burning influences during the campaign.

## 3 Results and discussion

During the FIREX-AQ campaign, we collected samples from five large wildfires with a wide range of ages (Fig. S2
and Table 1). TBs were found in all measured wildfires and agricultural smoke. We show the TB morphology, mixing states,
compositions, sizes, and abundances in Section 3.1 and the TB abundance and compositions from different smoke ages in
Section 3.2. TBs in a PyroCb event are provided in section 3.3. We discuss the possible TB formation processes in Section 3.4
and the possible implications of our results for climate impacts in Section 3.5.

### 3.1 Occurrence of tarballs

### 3.1.1 Morphology and mixing states of the tarballs and other organic particles

TBs retain the spherical shape with minimal deformation on the substrate. Their spherical shapes indicate that they
had solidified in the atmosphere or had a high enough viscosity to retain their sphericity on the substrate, even after being
sampled by jet stream impaction. Note that TBs may experience slight deformation on substrates; however, they generally
exhibit their maximum diameter near their center with minimal deformations on the contact surface with the substrate (Fig. 1).
Most TBs have negligible coatings, except for those in clouds (Fig. 2c–d). These spherical TB shapes and negligible coatings
are consistent with those found in other samples from various biomass burnings (Pósfai et al., 2004; Hand et al., 2005; Adachi
et al., 2019; Yuan et al., 2020), although we found exceptions, as shown in Section 3.3.



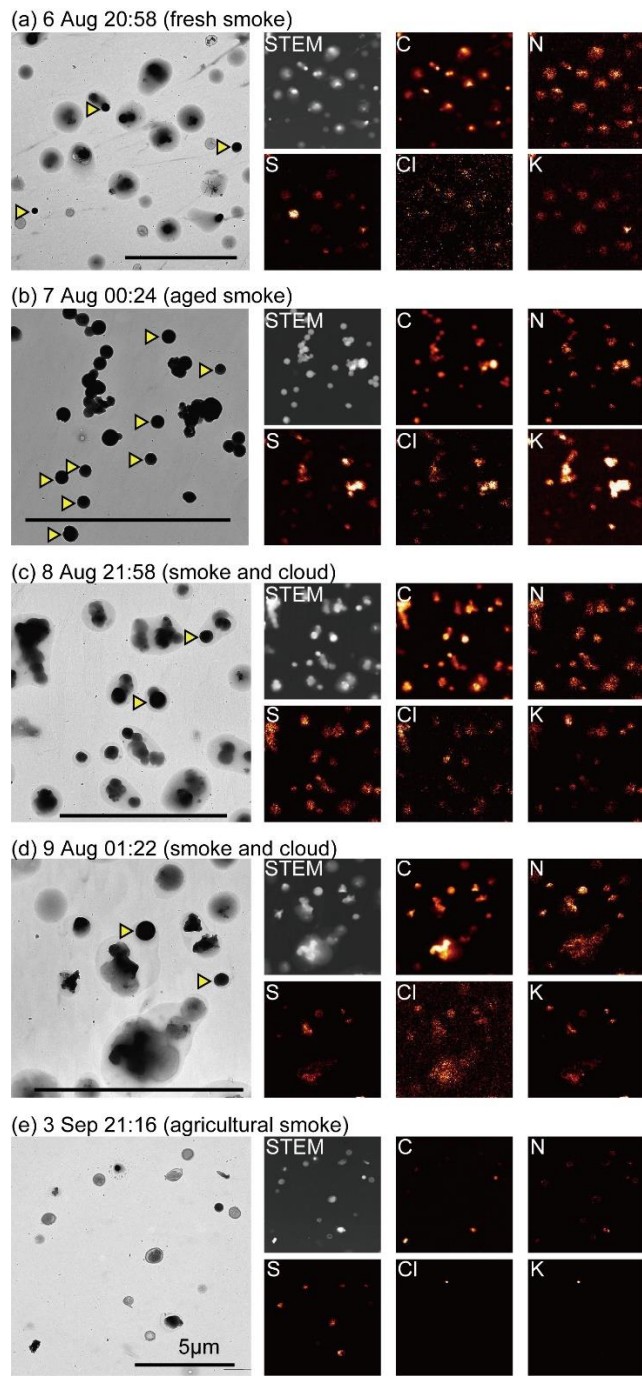

**Figure 2. Mapping images of various TEM samples. (a) Fresh (~0.2 h from the emission) and (b) aged (~3.6 h from the emission) samples from the Horsefly wildfire. (c) Wildfire sample within the cloud. (d) Williams Flats wildfire sample within PyroCb. (e) Agricultural smoke sample. Triangles indicate TBs. Scale bars indicate 5 μm.**






Unlike TBs, other organic particles extensively deform on substrates due to their low viscosity, resulting in flattened or dome-shaped configurations (Fig. 1d). Such flattened shapes can be distinguished by a short electron beam passing through the particles, which appear brighter than TBs in the TEM image and can also be recognized in the tilted images of representative samples (Fig. 1).

TBs are typically observed as individual particles without mixing or coagulation with other aerosol particles (Pósfai et al., 2004). However, in some cases, TBs may occur in clusters or coagulation forms (Chakrabarty et al., 2006; Girotto et al., 2018). Our samples with relatively high TB number fractions showed TB coagulation (Fig. S3). Approximately 50% of samples with a TB number fraction between 5% and 10% exhibited one or more aggregated TBs in the representative TEM images (Fig. S3). These proportions increased in samples with higher TB number fractions; more than 80% of samples with
TB number fractions of >25% included aggregated TBs (Fig. S3). Most aggregated TBs comprise several or more primary TBs. In some samples, aggregated TBs comprised tens of TBs (Fig. 3a), although the number of primary TBs is typically lower than that for soot particles, which can contain hundreds of primary particles (Buseck et al., 2014). Although the precise TB aggregation mechanism remains unclear, we hypothesize that TB aggregates form when 1) TB remains solid or highly viscous enough to maintain its spherical shape upon colliding in the atmosphere and 2) the TB concentration is sufficiently high to
cause coagulation between particles. By measuring a wide range of biomass burning samples, we found that TBs were also occasionally mixed with other particles such as soot (Fig. 3b), ash particles containing Ca and Mg (Adachi et al., 2022a) (Fig. 3c), and, infrequently, brochosomes derived from leafhoppers (Fig. 3d) (Wittmaack, 2005; Adachi and Buseck, 2013).



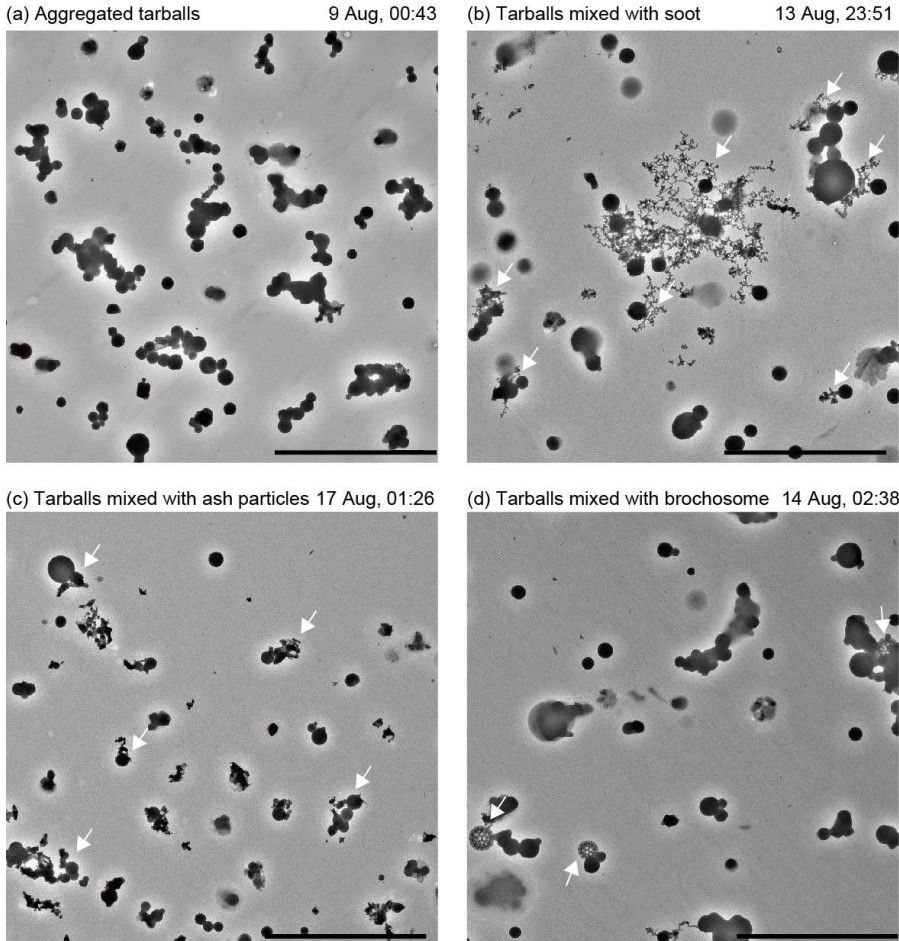

**Figure 3. TEM images of coagulated TBs. (a) Aggregated TBs from the Williams Flats fire. (b) TBs coagulated with soot particles from the Castle fire. Arrows indicate soot particles mixed with TBs. (c) TBs coagulated with ash particles from the Sheridan wildfire. Arrows indicate ash particles mixed with TBs. (d) TBs coagulated with brochosomes from the Castle wildfire. Arrows indicate brochosomes. Scale bars indicate 5 μm.**

### 3.1.2 Tarball compositions

TBs consist mainly of carbon ($89 \pm 4$ wt% on average) and oxygen ($8 \pm 2$ wt%). Minor constituents include nitrogen, silicon, sulfur, and potassium ($>0.1$ wt%), as well as trace amounts of chlorine and iron ($>0.05$ wt%). These elements detected in TBs are consistent with those reported in previous studies (Pósfai et al., 2004; Hand et al., 2005; Adachi and Buseck, 2011; Chen et al., 2017; Adachi et al., 2019; Yuan et al., 2020). It is possible that TB precursors are primary carbonaceous materials with low viscosity and are homogeneously mixed with inorganic compounds, including KCl, $K_2SO_4$, silicon, and nitrate during the TB formation in the smoke.





### 3.1.3 Tarball sizes

The modal sizes of TBs and carbonaceous particles (non-TBs) in the current study were 0.38 ± 0.08 and 0.49 ± 0.18 µm, respectively (Fig. 4), which were within the range measured by on-line instruments during the FIREX-AQ campaign (Moore et al., 2021). TBs had smaller and narrower size distributions than carbonaceous particles. One reason why
carbonaceous particles exhibit a wider size distribution is their deformation on the substrate, which depends on their viscosity and increases their apparent sizes, as represented by the area-equivalent diameters. In contrast, TBs deform less on the substrate and tend to have geometric sizes similar to those in the atmosphere. It should be noted that some TBs are not perfectly spherical and are slightly elongated when viewed from tilted images, i.e., the aspect ratio is ~1.16 from views in the 60-degree tilted TEM images toward the horizontal axis (Fig. 1d). Thus, their size in the atmosphere before sampling could be slightly smaller
than that shown in the current study.

The size of TBs in the FIREX-AQ samples was relatively large compared to that found in previous studies, which showed that TBs typically range between 0.2 and 0.3 µm (Pósfai et al., 2004; Hand et al., 2005; China et al., 2013; Girotto et al., 2018; Adachi et al., 2019; Yoshizue et al., 2020; Yuan et al., 2020), although some studies have also reported larger TBs, measuring over 0.4 µm (Chakrabarty et al., 2006; Tivanski et al., 2007; Fu et al., 2012). Although we could not identify the
specific reasons, possible reasons for the large TB sizes in the current study may include differences in the formation processes, such as fuels, burning conditions, and aging, as well as differences in the sampling and measurement conditions.

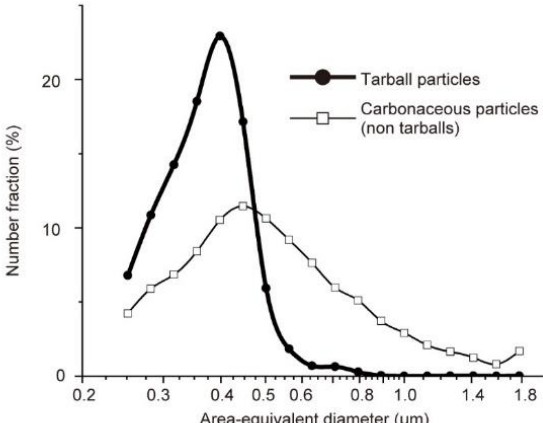

**Figure 4. Size distributions of TBs and non-TB carbonaceous particles. The particle size distribution peaks of TB and carbonaceous particles using the Gaussian fit with one sigma value are 0.38 ± 0.08 µm and 0.49±0.18 µm, respectively.**
**Bin sizes are shown in log scale: <0.28, 0.28–0.32, 0.32–0.35, 0.35–0.40, 0.40–0.45, 0.45–0.50, 0.50–0.56, 0.56–0.63, 0.63–0.71, 0.71–0.79, 0.79–0.89, 0.89–1.00, 1.00–1.12, 1.12–1.26, 1.26–1.41, 1.41–1.58, 1.58–1.78, and >1.78 µm. These sizes are all area-equivalent diameters measured from STEM images. n = 4567 and 27307 for TBs and the carbonaceous particles, respectively.**



### 3.1.4 Tarball abundance for each smoke

The TB number fractions for each flight varied between 1% and 20% (10% ± 1% on average) (Fig. 5). The estimated TB mass fractions for each flight were between 1% and 25% with an average value of 10% ± 2%. The estimated $TB_{mc}$ (µg m$^{-3}$) and TB enhancement ratios also varied depending on flights ranging from 2 to 35 µg m$^{-3}$ (10.1 ± 4.6 µg m$^{-3}$ on average) and from 0.001 to 0.03 (0.01 ± 0.002 on average), respectively. The TB abundances and enhancement ratios from the same wildfires, but on different sampling days, were similar (e.g., Williams, Castle, and Sheridan Fires), indicating that the type of wildfire

and its fuel sources largely influence the TB emission. In addition, the TB abundance differs significantly depending on the age of the sample, which is discussed in Section 3.2.

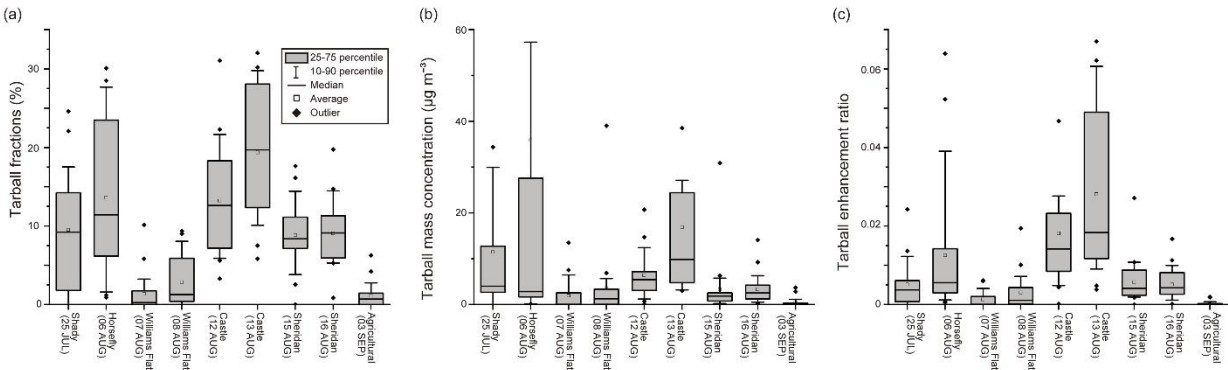

**Figure 5. Number fractions, mass concentrations, and enhancement ratios ($TB_{mc}/d$CO) of the TBs within the TEM samples for each flight**.

       Agricultural fires emit fewer TBs than wildfires. The TB number fraction, $TB_{mc}$, and enhancement ratio are 1% ± 1%, 0.5 ± 0.4 µg m$^{-3}$, and 0.002 ± 0.001, respectively. A possible explanation is that differences in their fuel compositions,

i.e., woody biomass fuels contain more lignin and cellulose than agricultural biofuels (Travis et al., 2023). Additionally, our TEM study during the FIREX-AQ campaign revealed that inorganic matter (sulfate and K-bearing particles) was more prevalent in agricultural fires than in wildfires (Fig. 6). The differences in their dominant fuel types and burning conditions likely contributed to lower emissions of TBs, compared with wildfires.

### 3.1.5 Tarball number fractions for each size range

The size-dependent number fractions of aerosol particle types are shown for each flight (Fig. 6). The figures are essentially the same as Fig. S8 in Adachi et al. (2022a), with the addition of the TB fractions in this study (highlighted in yellow in Fig. 6). Unlike other types of aerosols widely distributed across all size ranges, TBs are mainly found in the smaller





size ranges (0.25–0.50 µm). The TB fractions are ~20% for wildfires within these small size ranges (<0.5 µm) and are the second largest fraction next to the carbonaceous particle fraction at the size ranges.

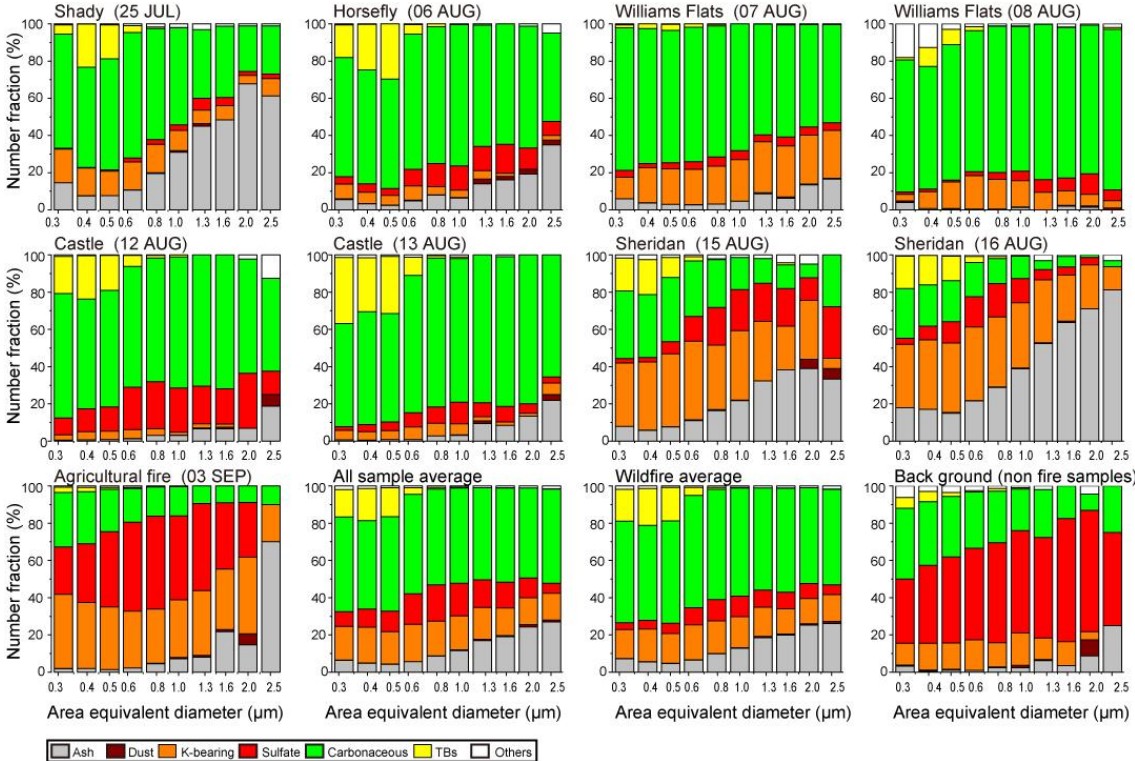


**Figure 6. Size-dependent number fractions of aerosol particles collected during the FIREX-AQ campaign. The size bins are 0.25–0.32, 0.32–0.40, 0.40–0.50, 0.50–0.63, 0.63–0.79, 0.79–1.00, 1.00–1.26, 1.26–1.58, 1.58–2.00, and >2.00 µm.**

## 3.2 Tarball abundance and composition in samples with different smoke ages

A series of TEM images demonstrates the TB formation process following an increase in smoke age during the Horsefly fire event on August 6[th] (Fig. 7). When the samples were collected from the fresh smoke (0.2 h), other organic particles with low viscosity dominated. They spread over the substrate and showed weak contrast in the TEM image (Fig. 7a). Some contained inclusions of either inorganic or organic matter. Some TBs with relatively small sizes were also observed (TB number fraction: 7%). The percentage of TB fractions slightly increased to 11% when samples were taken from smoke that

had aged for 1.3 h. However, the sample still contained numerous low-viscosity organic matter (Fig. 7b). As the aging process continued (~2 h), the TB fraction increased to 18% and 23%, resulting in higher apparent organic viscosity with darker contrast in the TEM image (Fig. 7c–d). The most aged samples (>3 h) had a higher number of TBs (29%) and high-viscosity organic matter (Fig. 7f). These aged samples also contained aggregated TBs.



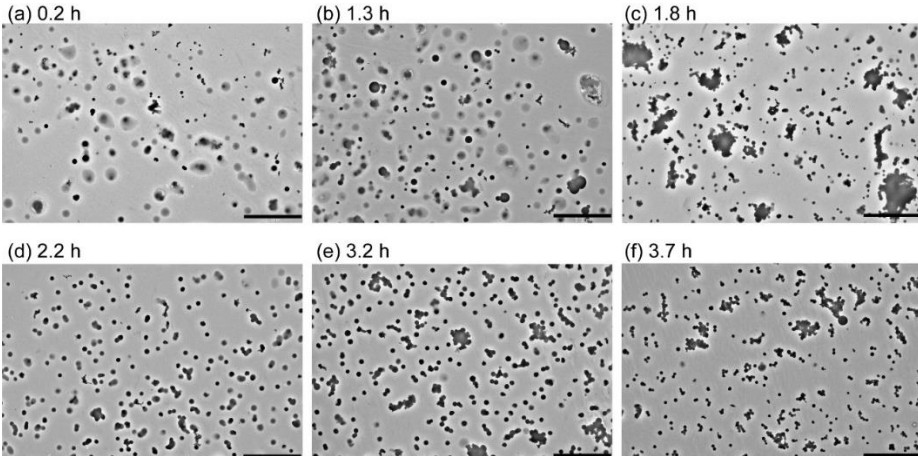

**Figure 7. TEM images with different aging and TB fractions in the Horsefly wildfire. Samples were collected on (a) 6 Aug (20:58), (b) 6 Aug (21:19), (c) 6 Aug (23:27), (d) 6 Aug (23:37), (e) 7 Aug (00:11), and (f) 7 Aug (00:22) in UTC. The estimated aging hours (h) from the emissions are shown in each TEM image. The TB number fractions of the corresponding samples were 7%, 11%, 18%, 23%, 24%, and 29%, respectively. Scale bars indicate 5 μm.**

TB number fractions are plotted along with their estimated aging time to compare TB formation in various types of smoke as a function of smoke aging (Fig. 8). Although these plots show large variability, they showed positive correlations for samples younger than ~5 h from the emissions. In particular, the samples from the Horsefly (Fig. 8b) and Williams Flat (Figs. 8c and 8d) fires showed reasonable correlations between TB fractions and the smoke age ($R^2$ = 0.38, 0.38, and 0.47, respectively). In all samples with the same smoke age, averaged TB number fractions increased from 5% to 15% as the aging proceeded up to 5 h, followed by a subsequent decrease in TB fraction (Fig. 9a). For reference, the $TB_{mc}$ and enhancement ratios at different smoke age are also shown in Fig. 9b and 9c. These values also support the increasing trend up to 5 h from the emissions and the subsequent decreases in the aged samples (> 5 h). We interpret the results that TB formation was almost complete around 5 h after the emissions. In samples aged >5 h, the TB abundances started to decrease, likely due to dilution, mixing with other aerosol particles from non-biomass burning sources, and TB removal from the atmosphere.



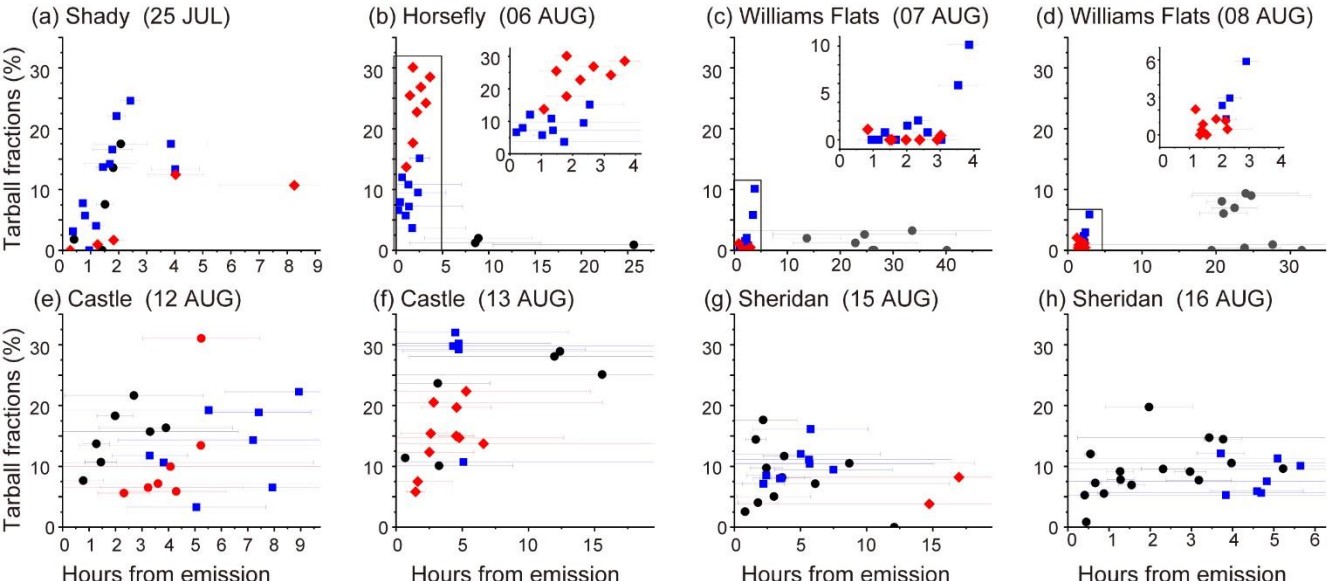

**Figure 8. TB number fractions and hours from the emissions for each wildfire (a-h). Different symbols (red diamonds, blue squares, and black circles) in each panel indicate different repeated flight patterns for each wildfire smoke. Correlation coefficient values ($R^2$) from (a) to (h) are 0.15, 0.38*, 0.38*, 0.47*, 0.07, 0.26*, 0.06, and 0.04, respectively (*significance level, $p < 0.05$). The $R^2$ values for (b), (c), and (d) were obtained from two repeated flights of less than 4 h, as shown in the upper right of each panel.**

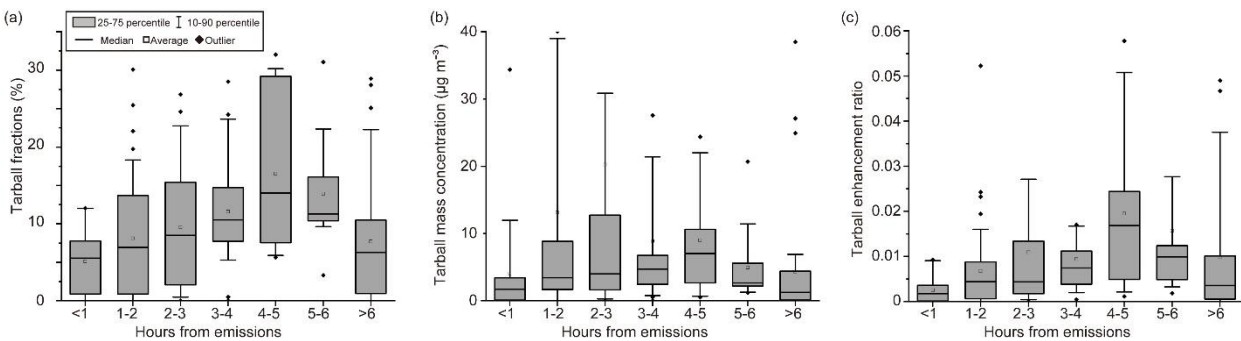

**Figure 9. Hourly averaged (a) number fractions, (b) mass concentration ($TB_{mc}$), and (c) enhancement ratios ($TB_{mc}/dCO$) of TB within all wildfire smoke samples. n = 18, 41, 29, 25, 14, 13, and 34 from <1 h to >6 h bins, respectively.**

As they aged for up to 6 h, non-sulfate N increased in both TBs and other organic particles (Figs. 10a and 10b). Non-sulfate O in TBs slightly increased with age up to 6 h, but its increase was less clear than that of N (Fig. 10c). Samples aged <1 h have a high average O concentration (~7 wt%) with large 25–75 percentile ranges because there are only three



samples, resulting in a large uncertainty. As they aged, no obvious increase in O was observed in carbonaceous particles (Fig. 10d).

The results of increasing N in TBs as smoke age increases are consistent with those in BBOP (Adachi et al., 2019). It was hypothesized that the increase in organic matter viscosity during TB formation in biomass burning plumes proceeds by the addition of N and O in initially low-viscosity organic particles by the formation of, for example, carboxylic acid and organic nitrogen compounds within several hours of emission by measuring their individual particle compositions using electron energy loss spectrometry and scanning transmission X-ray spectroscopy (Adachi et al., 2019). The current results support this hypothesis for N by analyzing samples that are more aged than those in BBOP (Kleinman et al., 2020). The increase in O with increasing smoke age was not explicitly observed in the current study, and further observations are needed to conclude it. Increases in N and O in TBs can both decrease (bleaching by oxidation) and increase (formation of organic nitrate) light absorption (Li et al., 2019).

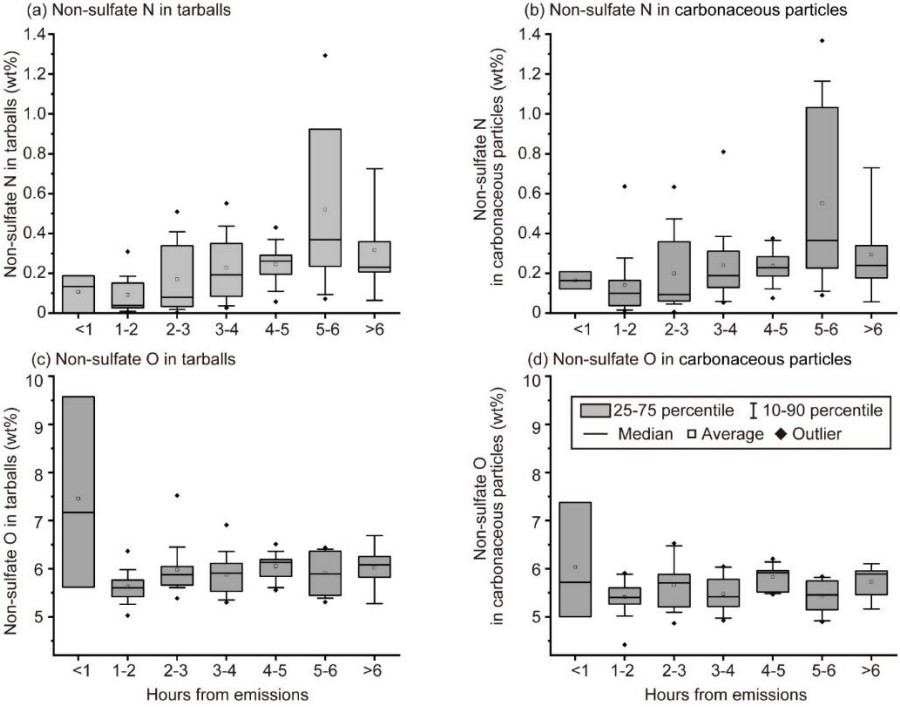

**Figure 10. Averaged nitrogen and oxygen weight percent within TBs and carbonaceous particles (non-TBs) for samples. Non-sulfate N wt% are shown in (a) TBs and (b) carbonaceous particles. Non-sulfate O wt% are shown in (c) TBs and (d) carbonaceous particles. These plots used averaged values for the only samples having TB number fraction >10% as the samples with few TBs could include background air with well-aged TBs. n = 3, 13, 12, 14, 10, 11, and 9 from <1 h to >6 h bins, respectively.**



### 3.3 Tarballs in pyrocumulonimbus (PyroCb)

During the FIREX-AQ campaign, we had a chance to observe a pyrocumulonimbus (PyroCb) event, which is a thunderstorm that forms from biomass-burning smoke (Peterson et al., 2022; Warneke et al., 2023). The PyroCb occurred in the Williams Flats wildfire smoke that ascended to an altitude of approximately 10 km, which is the upper troposphere and lower stratosphere, and formed ice-phase clouds or mixed-phase clouds of water droplets and ice crystals (Peterson et al., 2022).

The average TB fractions in the Williams Flats smoke were relatively small (1%–3%; Fig. 5 and Table 1) and showed an increase in TB fractions as smoke age increased, similar to other biomass burning smoke plumes (Fig. 8). Representative TEM images of the Williams Flats smoke samples also showed that the viscosity of organic particles was low in the fresh samples (Fig. 11e and 11f) and that the fractions of high-viscosity organic particles increased in the aged samples (Fig. 11a and 11b). The samples collected at the top of PyroCb (~9 km) were dominated by other organic particles with low viscosity and had a small number of TBs (<2% in number fractions) (Fig. 11c and 11d). In the aged samples (Fig. 11a and 11b), a large fraction of high-viscosity organic particles appeared dark and nonspherical in the TEM images.

Some TBs and other organic particles collected in clouds in the Williams Flats TEM samples had thin layers or coatings (e.g., Figs. 2c, 2d, 11c, and 11d). Such coatings are traces of water mixed with water-soluble matter (Semeniuk et al., 2006) that contain, for example, N, S, and Cl (Fig. 2). Despite water evaporation after sampling, residuals may still surround the water-insoluble cores on the substrate. In the atmosphere, while the smoke ascends, the temperature decreases, and TBs and other aerosol particles can become CCNs and even INPs depending on the temperature, relative humidity, and properties of the aerosol particles (Petters et al., 2009; Jahl et al., 2021). As a result, water condensation or ice development occurred on the particle surface. Nonetheless, some TBs did not show any trace of water, indicating that they were relatively less hygroscopic than other particles (e.g., Fig. 11d). Although ice crystals may have been present in the PyroCb at the sampling altitude (~9 km), they are typically larger than our cutoff sizes (Peterson et al., 2022), making it less probable that the collected particles contained ice crystals.

TBs with a trace of water retained their spherical shapes (Fig. 2d). These results indicate that although some TBs became CCN, they do not comprise highly hygroscopic substances and do not readily dissolve in water. This result is inconsistent with that of Hand et al. (2005), who showed the hygroscopic properties of TBs, but agrees with Semeniuk et al. (2006) and Adachi and Buseck (2011), who showed less hygroscopicity of TBs when they were exposed to high relative humidity in the environmental TEM chamber. In conclusion, TBs can serve as CCNs but exhibit low efficiency.



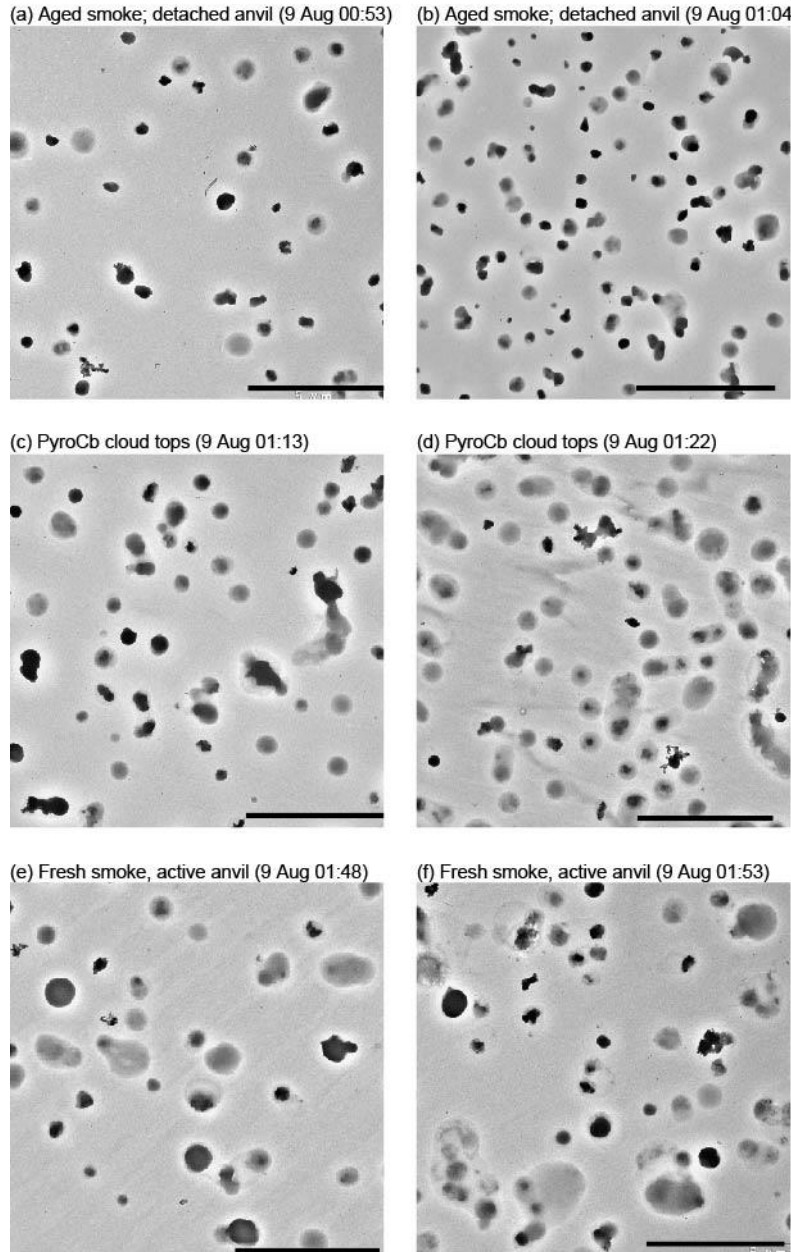

**Figure 11. TEM images of aerosol particles within PyroCb from the Williams Flats wildfire. These samples were collected from the same flight with different agings and cloud positions. The sample altitudes were between 7 and 9 km.**

**These samples correspond to transect numbers 1 (a), 2 (b), 4 (c), 5 (d), 8 (e), and 9 (f) in Fig. 4 by Peterson et al. (2022), and the cloud descriptions are also followed by their study. Scale bars indicate 5 μm.**



### 3.4 TB formation

Previous studies have shown TB formation in wildfire smoke, but they focused on moderately aged (up to 3 h)
smoke samples (Sedlacek et al., 2018; Adachi et al., 2019). This study showed a similar processed primary particle formation
as that shown in Sedlacek et al. (2018) with continued TB formation in more aged samples (~5 h) (Fig. 9). Formation rates
and abundance varied depending on the type of biomass burning smoke. Some had an unclear relationship between TB
fractions and smoke age (e.g., Sheridan) and contained only a few TBs (e.g., Williams Flats and most agricultural fires) (Fig.
8 and Table 1). These results suggest that other parameters, in addition to aging, such as fuel type and burning conditions,
could also largely contribute to TB formation. Increases in N were observed with increasing age up to 6 h (Fig. 10), suggesting
the formation of organic nitrogen compounds. This process is likely related to TB formation (Adachi et al., 2019) or proceeded
concurrently with it. Nonetheless, the primary factor in TB formation is increasing viscosity with increasing age. In addition
to TB formation, the increase in viscosity of organic particles has implications for gas-particle reactions, affecting reaction
degree and speed (Liu et al., 2018; Reid et al., 2018). In future studies, the factors that lead to the increasing viscosity of
420 organic matter, such as addition of specific functional groups (Reid et al., 2018) and water loss, should be measured.

### 3.5 Implications for climate-relevant properties

TBs are known to be light-absorbing carbon (brown carbon) (Chakrabarty et al., 2010; Sedlacek et al., 2018). During
FIREX-AQ, Chakrabarty et al. (2023) showed that TBs contributed three-quarters of the short visible light absorption and half
of the long visible light absorption within samples collected from ground samplings. The evaluated imaginary refractive index
$k$ of the TB from Shady fire ranged between $0.06 \pm 0.03$ and $0.13 \pm 0.04$ at a wavelength of 550 nm depending on the burning
conditions. They also indicated that TBs are water-insoluble, resist daytime photobleaching, and enhance their light absorption
with night-time atmospheric processing. As part of our samples were collected from the same wildfire (e.g., Shady), we assume
that they could have similar optical properties.

The degrees of attachment or coating on TBs (i.e., mixing states) affect their hygroscopicity, optical properties, and
430 influences on human health. We found that TBs are sometimes coagulated or attached soot particles (Fig. 3b), resulting in
changing their light absorption properties (Lack et al., 2012; Adachi and Buseck, 2013; Saleh et al., 2014). However, TBs did
not embed or internally mix with soot, implying the lens effects enhancing the soot absorption (Bond et al., 2006) can be
ignored. Our TBs frequently showed aggregated shapes (Fig. S3). Their aggregated shapes influence their optical properties,
potentially increases their single-scattering albedo by up to 41% at a wavelength of 550 nm compared with individual TBs
(Girotto et al., 2018).

Large biomass burning smoke is rapidly transported to the upper troposphere and lower stratosphere due to the heat
produced by the wildfire, forming PyroCb (Fromm et al., 2019; Peterson et al., 2022). Such PyroCb injects significant amounts
of biomass burning emissions into the upper troposphere and lower stratosphere. Our measurements found that TBs can also

be one of the aerosol species in PyroCb transported to the high atmosphere. Such stratospheric aerosol particles suspend for
440     months, affecting the climate (Katich et al., 2023).

Overall, the influence of TBs on climate depends on their abundance, refractive index, mixing states, morphology,
and atmospheric lifetime. These factors can have both positive and negative effects. This study revealed that TBs are not simple
inert spherical organic particles but have various appearances that need to be considered regarding their climate influences.

## 4 Summary and conclusion

This study collected multiple biomass burning smoke and a wider range of aging than previous studies and detected
4567 TBs using a deep learning image analysis method. The number fractions, estimated mass concentrations, and
enhancement ratio of TBs increased within 5 h following emissions. Our findings provide further evidence regarding the TB
formation via increased viscosity of primary organic matter. They can coagulate with soot and ash particles and other TBs,
forming TB coagulations. These mixing states could influence their optical properties. Together with other biomass burning
emitted aerosol particles, TBs can be transported to the upper troposphere and lower stratosphere through PyroCb and could
survive for months, potentially altering solar radiation. Although TBs can condense water under some conditions, they retained
their spherical shapes in the droplets, suggesting that they are sparingly soluble in water. Although TBs have been presumed
to be spherical particles that do not mix with other particles, this study found a wide range of TB mixing states that need to be
considered when evaluating their climate influences.

## Data availability


FIREX-AQ data are available at https://www-air.larc.nasa.gov/cgi-bin/ArcView/firexaq. STEM-EDS data for all individual
particles and those for the TEM sample average are available at https://doi.org/10.5281/zenodo.10751569.

## Author contributions

KA conducted the TEM analysis and data processing. KA, JED, and JMK set up and executed the TEM sampling. CDH
estimated the sample age. HG, PCJ, and JLJ conducted AMS measurements. JP measured CO concentrations. JPS and JC
supervised the campaign. KA prepared the manuscript with contributions from all coauthors.

## Competing interests

The authors declare that they have no conflict of interest.



## Acknowledgments

We acknowledge the science team members, supporting staff, and the pilots and flight staff of the research aircraft for FIREX-AQ (NASA DC8). KA thanks the Environmental Research and Technology Development Fund (JPMEERF20172003, JPMEERF20202003, JPMEERF20215003, and JPMEERF20232001) of the Environmental Restoration and Conservation Agency of Japan, the Global Environmental Research Coordination System from the Ministry of the Environment of Japan (MLIT1753 and MLIT2253), the Arctic Challenge for Sustainability II (ArCS II) (JPMXD1420318865),

and the Japan Society for the Promotion of Science (JSPS) KAKENHI program (grant numbers JP16K16188, JP19K21905, JP19H04259, JP23H03531, and JP23K28221) for financial support. HG, PCJ and JLJ acknowledge NASA Grants 80NSSC21K1451 and 80NSSC23K0828. We especially thank Eric Scheuer (deceased May 2022), who collected all TEM samples during the FIREX-AQ campaign.

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
