# Peer review of "Occurrence, abundance, and formation of atmospheric tarballs from a wide range of wildfires in the western US"

_EGUsphere, 2024_

## Author Comment (AC1)

Reviewer comments are shown in Bold.

Author replies are shown in normal font.

*Revised texts are shown in Italic.*

Reviewer #1:

**R1-1: Adachi et al. conducted a comprehensive transmission electron microscopy analysis to understand the tarball formation, concentration, and properties. Overall, the study is well organized, and the paper is well written. The results have a potentially high impact on understanding biomass burning aerosol climate effects. I have a few minor comments that I hope can be used to help the authors further improve the paper.**

AC1-1: We appreciate the supportive comments by reviewer #1.

**R1-2: The paper used the assumption that none TB carbonaceous aerosols' area-equivalent diameter is approximately twice the volume-equivalent diameter to estimate their volume. However, this study has acquired tilted images, which wight be a better and more direct way to estimate the volume of particles (Cheng et al., 2023, 2021)It might be worth discussing the uncertainty between these two methods and considering using the aspect ratio to calculate the volume in future studies.**

AC1-2: We thank Reviewer 1 for this suggestion. In the revised manuscript, we discuss the uncertainties and possible changes in particle shape. Please note that, although we obtained tilted TEM images using a high-tilt tomography TEM holder for selected samples, the holder cannot be used for the STEM-EDS analysis, and thus the tilted images were not available for most TEM samples. In future studies, we will consider having the tilted images. We mentioned the limitations in the manuscript as follows:

*(Section 2.4) An increase in relative humidity after sampling will change the particle shapes and could increase the apparent aerodynamic diameter (Cheng et al., 2021). Thus, we kept the samples dry during the campaign with a desiccant and stored them in a desiccator after the campaign.*

(section 2.6) *Conversely, non-TB particles were highly deformed on the substrates, and an assumption is needed to estimate their volume-equivalent diameters. Cheng et al. (2023) showed a method for estimating particle volume from tilted images and particle aspect ratio. Although we obtained tilted TEM images using a high-tilt tomography TEM holder for selected samples (Fig. 1), the holder cannot be used for the STEM-EDS analysis, and thus the tilted images were not available for most TEM samples.*

R1-3: I suggest adding a sentence to acknowledge the caveat that volatile and semivolatile species in the particles could be lost in the high-vacuum TEM chamber, causing shape deformation. Typically, the particle height will reduce, but the base will maintain the same area due to the fraction and adhere force.

AC1-3: We added the following discussion:

*(Section 2.4) Volatile and semi-volatile particles may be lost during sampling and in the vacuum TEM chamber, causing a shape deformation of such materials. The particle height on the substrate may decrease post-collection due to the loss of volatile materials, but the base will remain the same area due to the friction and adhesion forces and will retain its original area-equivalent diameter.*

R1-4: Could you provide any estimation of how much C and O could come from the formvar due to the penetration effect? Also, the C, N, and O from EDX are semiquantitative.

AC1-4: We added the following discussions:

*(Section 2.4) EDS signals of C and O can also originate from the carbon substrate. The influence of the substrate is large when measuring thin organic coatings, and their EDS intensities from the substrate can be up to half that of the organics (Adachi et al. 2016). However, thick carbonaceous particles, such as TBs and soot particles, have C signals ~7 times higher than the substrate (Adachi et al., 2016), making them clearly distinguishable from the substrate.*

*(Section 2.4) The uncertainty for light elements (N and O) was generally high and was evaluated as 5% or less (Adachi et al., 2019).*

R1-5: I suggest adding Si maps to figure 2 since you discussed Si in section 3.1.2.

AC1-5: We appreciate this suggestion. The Si mappings are shown below. The mappings show that some TBs contain Si (e.g., panel b bottom). However, we hesitate to include the Si mapping in the main text because the mapping images show an artificial effect of Si around the edges of the TEM images. This artificial Si effect was caused by the deposition of Si-containing molecules in the TEM chamber when an electron beam was exposed for a long time (almost one hour) to obtain the mapping images. Such an artificial effect does not affect the normal STEM-EDS analysis with short exposure time (20 seconds). Instead of showing the Si mapping images in the main text, we prefer to show them only in this reply file, which can also be accessed by the public with this context that we think is too tangential for the main text.

[Figure]

(a) 6 Aug 20:58 (fresh smoke)

(b) 7 Aug 00:24 (aged smoke)

(c) 8 Aug 21:58 (smoke and cloud)

(d) 9 Aug 01:22 (smoke and cloud)

(e) 3 Sep 21:16 (agricultural smoke)

**R1-6: Would it be a potential reason that the increase fraction of TBs with atmospheric aging time is due to the sink of volatile species due to rapid dilution with air and rapid decomposition?**

AC1-6: Although the proposed process may contribute to the increase in TB number fractions, the estimated TB mass concentration and enhancement ratios also increased (Fig. 9), suggesting that TB mass increased independently of the decrease in volatile species. In addition, we could not confirm the evidence of the decrease of volatile species. Thus, we could not confirm the suggested point. Please see AC1-3 for relevant changes in response to this point.

**R1-7: Do you think that those inorganics in the carbonaceous could be formed during the condensation phase of the flame where carbonaceous species in TB condensed on the inorganic compounds or the inorganic compounds diffused into TBs?**

AC1-7: We think that those inorganic species in the low-viscous organic particles were formed during the condensation phase of the flame as they were found in relatively fresh smoke samples (e.g., Fig. 7a), although we do not have direct evidence to conclude this in the current manuscript.

**R1-8: Figure S3 caption is unclear to me. Is the y-axis the fraction of aggregated TBs in the sample? It is not very clear to me what is the number of samples containing TBs? Is this the total number of particles analyzed in that sample? How many TEM grids have you analyzed?**

AC1-8: The Y-axis indicates the number fraction of samples having aggregated TBs over all analyzed samples. The samples containing TBs are defined as those with any aggregated TBs in the representative TEM images. This is the total number of TEM samples that we analyzed. In total, 221 TEM samples were analyzed (Table 1). We revised the caption to explain above as follows:

*(Caption in Figure S3) Fractions of samples having aggregated TBs for samples with various TB number fractions. Samples were classified into each bin based on their TB number fraction (X-axis). The samples with any aggregated TBs in the representative TEM images were defined as the samples having aggregated TBs ($Sample_{aggregatedTBs}$). The number fraction of $Sample_{aggregatedTBs}$ over all sample numbers in each bin is shown on the Y-axis. The number of samples containing TBs was 179, while that of $Sample_{aggregatedTBs}$ were 84. n = 54, 50, 36, 16, 10, and 13 for the bins of samples with TB fractions of 0%–5%, 5%–10%, 10%–15%, 15%–20%, 20%–25%, and >25%, respectively.*

R1-9: Mathai et al., 2023 show that TBs with and without inclusion have different optical properties (Mathai et al., 2023). It will be nice to link that study with your findings.

AC1-9: We added the reference and revised the text based on their results:

*(Section 1) Further TB measurements have also revealed that they have light-absorbing optical properties (Hand et al., 2005; Chakrabarty et al., 2010; Sedlacek, et al., 2018; Mathai et al., 2023).*

*(section 3.1.2) It is possible that TB precursors are primary carbonaceous materials with low viscosity and are inhomogeneously and homogeneously mixed with inorganic compounds (Mathai et al., 2023), including KCl, K₂SO₄, silicon, and nitrate during the TB formation in the smoke.*

*(section 3.5) TBs are known to be light-absorbing carbon (brown carbon) (Chakrabarty et al., 2010; Sedlacek et al., 2018; Mathai et al., 2023).*

**Reviewer #2:**

R2-1: The paper reports the abundance of TBs from several fires in the US. TBs can have an important role in the climate impact of fires and, therefore, the work is worth publication. The methods used seem sound and the exposition is mostly clear. The results are very interesting. I have only a few comments to follow.

AC2-1: We appreciate Reviewer #2's helpful comments in improving the manuscript.

**General comments**

R2-2: - I am not clear about the inlet used and the impact that the aircraft speed might have had on the samples, especially the size cut of the impactor. It would be helpful if the authors could comment on this aspect as that might affect the interpretation of their size results.

AC2-2: We used a common aerosol inlet operated by the NASA Langley Aerosol Research Group (LARGE) on the DC-8. The inlet has been used in various previous campaigns on DC-8 and has been described in details (e.g., McNaughton et al., 2007; Moore et al., 2021; Brock et al., 2019). The 50% passing efficiency of aerosol particles are well larger than 1μm, depending on aircraft speed and particle density, and are greater than the upper cutoff size of our TEM sampler (700 nm). Therefore, the aircraft inlet cutoff size will not affect the particle sizes collected for the TEM analysis. We added this description in the revised text:

*(section 2.3) This study primarily used the Formvar substrates to observe TB shapes with a flat background image. The collected particles have small and large 50% cutoff sizes of aerodynamic diameters of 100 and 700 nm, respectively. We used a common aerosol inlet*

*operated by the NASA Langley Aerosol Research Group (LARGE) on the DC-8 (McNaughton et al., 2007; Moore et al., 2021; Brock et al., 2019). The upper cutoff aerodynamic diameter for 50% passing efficiency of aerosol particles by the inlet depends on aircraft speed and particle density (McNaughton et al., 2007), but is much larger than that of the TEM sampler, resulting in minimal effect on the TEM samples.*

**R2-3: -   In some cases, I found some details to be glossed over a bit too much. Some more on this in the specific comments next.**

AC2-3: We revised the manuscript to clarify the suggested points.

**Specific comments**

**R2-4: -   Line 22: This is a not too critical and arguable terminology suggestion. "As the samples aged⋯" the authors probably refer to the particles aging, not the sample aging, correct? The samples have been analyzed probably a long time after collection, so the time the authors refer to is for the particles at the time of sampling, not for the sample itself. Same in line 24 and section 2.2.**

AC2-4: Yes, we discuss particle aging in the smoke. "Sample age" was replaced with "smoke age" throughout the manuscript.

**R2-5: -   Line 24: If the fraction (not the concentration itself) of TBs is decreasing that should not be due to dilution alone. Does the decrease in fraction indicate some sort of selective removal?**

AC2-5: We think a mixing with background aerosol particles from non-biomass burning sources contributes to the decrease of TB fraction after substantial dilution. In addition, as the TB mass fraction and enhancement ratios also decreased (Fig. 9), the dilution and removal processes were also possible. We explained the point in the main text. In the revised abstract, we deleted the relevant sentence as the word count exceeds the limit (250words) when we explain these three possibilities. Please also see AC2-24 in response to this point.

*(section 3.2) In smoke aged >5 h, the TB abundances started to decrease, likely due to mixing with other aerosol particles from non-biomass burning sources, dilution, and TB removal from the atmosphere, which could mainly contribute to the decrease in TB number fractions, mass concentrations, and enhancement ratios, respectively.*

**R2-6: -   Line 99: Perhaps "measured" should be "analyzed"**

AC2-6: Revised.

**R2-7: -   Table 1: Would it be possible for the authors to estimate statistical errors on the fractions? One could at least use counting statistics to estimate the potential error.**

AC2-7: We added the 95% confidence intervals in Table 1.

**R2-8: -   Line 107: What are "archived winds"? Can you archive wind?**

AC2-8: "Archived winds" should be "archived wind speed data." We revised the text.

**R2-9: -   Line 114: Could the authors explain why they favored the Formvar substrates? This becomes evident only later in the paper.**

AC2-9: We used the Formvar substrate because it has a uniform background image and makes the image analyses easier than the lacey carbon substrate. We revised the sentence as follows:

*(section 2.3) This study primarily used the Formvar substrates to observe TB shapes with a flat background image.*

**R2-10: -   Section 2.4. How much time passed between sampling and analysis? Does that potentially impact the interpretation of the results?**

AC2-10: Shortly after the end of the campaign (September, 2019), we performed a preliminary analysis (TEM image measurements for all 221 samples) until December, 2019 (~20 TEM samples per a week). After the TEM image measurements, we analyzed ~5 TEM samples per week for the STEM-EDS measurements until September, 2020. We measured and compared the TEM images from the preliminary analysis and the STEM-EDS analysis and found no differences between them. The possible particle change could be the loss of water and volatile materials, which would be lost immediately after sampling. A change of relative humidity could also change particle shapes, and we thus kept the samples in a dry condition.

*(section 2.4) An increase in relative humidity after sampling will change the particle shapes and could increase the apparent aerodynamic diameter (Cheng et al., 2021). Thus, we kept the samples dry during the campaign with a desiccant and stored them in a desiccator after the campaign. Comparison of early TEM images with STEM-EDS performed up to a year later suggest that the samples were stable in storage over the time period of post-collection analysis.*

**R2-11: -   Line 127: Would that ratio depend on the viscosity of the particle? Same for line 197.**

AC2-11: Yes, it should be viscosity dependent. Ideally, we should know their viscosity particle by particle, but this is not realistic. Instead, we simply assume that the area-equivalent

diameters are approximately twice the volume-equivalent diameters (Zhang et al., 2020). *(section 2.6) Although the estimated values include significant uncertainties originating from, for example, particle collection efficiency (e.g., bouncing effects (Bateman et al., 2017) and loss of volatile particles) and particle viscosity, the attempt will provide an idea of how much TB particles are emitted from biomass burning smoke and the changes that occur with aging. (section 2.6) Conversely, non-TB particles were highly deformed on the substrates, and an assumption is needed to estimate their volume-equivalent diameters. Cheng et al. (2023) showed a method for estimating particle volume from tilted images and particle aspect ratio. Although we obtained tilted TEM images using a high-tilt tomography TEM holder for selected samples (Fig. 1), the holder cannot be used for the STEM-EDS analysis, and thus the tilted images were not available for most TEM samples. Instead, we assumed that the area-equivalent diameters of non-TB particles became two times larger than their volume-equivalent diameters (Zhang et al., 2020).*

**R2-12: - Paragraph starting at line 152: It would be useful to provide some more detail on the deep learning approach.**

AC2-12: We added following sentences to explain the deep learning approach:

*(section 2.5) Such deep learning image analysis can be achieved by first preparing training images (manually identified TB images) and then training the TEM model using the training images. After building the model, we applied it to all TEM images used for STEM-EDS analysis.*

**R2-13: - Lines 199-200: Units?**

AC2-13: The unit is $g/cm^3$. We add it in the revised manuscript.

**R2-14: - Line 223: How were TBs being sampled in clouds? Was a CVI being used, or are these interstitial TBs? Also, how was the coating being identified?**

AC2-14: We used the same inlet for all samples and did not use a CVI. Thus, they were interstitial TBs. Coating was identified as any apparent materials around the particles. We revised the sentence as follows:

*(section 3.1.1) Most TBs have negligible coatings, i.e., no apparent materials around the TBs, except for those collected in clouds (interstitial samples) (Fig. 2c–d).*

*(section 3.3) Some TBs and other organic particles collected in clouds (i.e., interstitial aerosol particles) in the Williams Flats TEM samples had thin layers or coatings (e.g., Figs. 2c, 2d, 11c, and 11d).*

**R2-15: -   Libe 232: "short electron beam" how short? And why?**

AC2-15: The electron beam path within a particle is equal to the particle thickness (commonly several tens nm). Thus, thinner particles have the shorter electron beam path, resulting in brighter TEM images. We revised the sentence:

*(section 3.1.1) Such flattened shapes can be distinguished by a short electron beam path through the thinner particles, which appear brighter than TBs in the TEM image and can also be recognized in the tilted images of representative samples (Fig. 1).*

**R2-16: -   Line 262: By "size" the authors indicate the diameter or the radius?**

AC2-16: The size is a diameter (area-equivalent diameter). We revised the sentence:

*(section 3.1.3) The modal sizes of TBs and carbonaceous particles (non-TBs) in the current study were 0.38 ± 0.08 and 0.49 ± 0.18 μm in area-equivalent diameters, respectively (Fig. 4), which were within the range measured by on-line instruments during the FIREX-AQ campaign (Moore et al., 2021).*

**R2-17: -   Line 268: Is this aspect ratio an average over all the TBs measured or only for those that showed some deformation?**

AC2-17: The aspect ratio is an average of those in the tilted images in Fig. 1.

*(section 3.1.3) It should be noted that some TBs are not perfectly spherical and are slightly elongated when viewed from tilted images, i.e., an average of the aspect ratio of representative TBs in Fig. 1 is ~1.16 from views in the 60-degree tilted TEM images toward the horizontal axis.*

**R2-18: -   Figure 4: There seems to be a smaller mode in the TBs size distribution. Can the authors comment on this smaller mode?**

AC2-18: We have discussed the reason as follows:

*(section 3.1.3) TBs had smaller and narrower size distributions than carbonaceous particles. One reason why carbonaceous particles exhibit a wider size distribution is their deformation on the substrate, which depends on their viscosity and increases their apparent sizes, as represented by the area-equivalent diameters. In contrast, TBs deform less on the substrate and tend to have geometric sizes similar to those in the atmosphere.*

**R2-19: -   Line 285: How was the 1% error being estimated? This also connects to one of my previous comments. Same for the 2% in the next line.**

AC2-19: They were 95% confidence interval values. We revised the sentence:

*(section 3.1.4) The TB number fractions for each flight varied between 1% and 20% (10% ±*

*1% on average with a 95% confidence interval) (Fig. 5).*

**R2-20: -   Lines 287-288: It would be useful if the authors would provide the method used to estimate the uncertainties in detail.**

AC2-20: They are 95% confidence interval values. Please also see our revision in AC2-19 in response to this point.

**R2-21: -   Line 288: Perhaps I missed it, but what does the enhancement ratio represent, and how is calculated? Why is important? The definition is reported in Figure 5 and perhaps was discussed earlier in the paper, but it might be good to elaborate some more on it.**

AC2-21: We described the method to obtain the mass concentration and enhancement ratio in section 2.6 and added the information here:

*(section 2.6) In addition to the TB number fractions, atmospheric TB mass concentration ($TB_{mc}$, µg m$^{-3}$) and TB enhancement ratios relative to carbon monoxide ($TB_{mc}/dCO$) (i.e., $TB_{mc}$ divided by dCO (ppb), which is $CO_{measured} - CO_{background}$) are useful for evaluating the TB climate effects (Yokelson et al., 2013).*

*(section 3.1.3) The estimated $TB_{mc}$ (µg m$^{-3}$) and TB enhancement ratios ($TB_{mc}/dCO$) also varied depending on flights ranging from 2 to 35 µg m$^{-3}$ (10.1 ± 4.6 µg m$^{-3}$ on average) and from 0.001 to 0.03 (0.01 ± 0.002 on average), respectively (see section 2.6 for the method to obtain the mass concentration and the enhancement ratio).*

**R2-22: -   Figure 5: The x-axis labels are very small and hard to read.**

AC2-22: We enlarged the font size from 8 to 10.

**R2-23: -   Figure 7: This is a beautiful result. Perhaps it would help to identify the TBs on the images to clearly underline the increase in fraction.**

AC2-23: Thank you for the comment. We added Figure S4 that marks TBs in red.

*(caption for Fig. 7) Detected TBs are shown in Fig. S4.*

**R2-24: -   Line 339: As mentioned earlier, I am not clear how a "fraction" would decrease with dilution alone. Was some selective scavenging going on, or was secondary organic aerosol forming and so dominating more the particle population?**

AC2-24: We hypothesize that mixing with other aerosol particles including both organic and inorganic particles from non-biomass burning sources contributed to the decrease in TB fractions.

*(section 3.2) We interpret the results that TB formation was almost complete around 5 h after*

*the emissions. In smoke aged >5 h, the TB abundances started to decrease, likely due to mixing with other aerosol particles from non-biomass burning sources, dilution, and TB removal from the atmosphere, which could mainly contribute to the decrease in TB number fractions, mass concentrations, and enhancement ratios, respectively.*

**R2-25: -  Figure 8: I am confused by the color/symbol explanation in the caption. For example, the authors mention red diamonds, blue squares, and black circles, but there are also red circles in panel (e). Probably those should also be diamonds. Also, I see that the colors/symbols represent different flight patterns, but what patterns do the authors refer to?**

AC2-25: Thank you for pointing it out. We revised the figure. They indicate different repeated transect flight pattern described in Warneke et al. (2023). We revised the caption:

*(caption of Fig. 8) Different symbols (red diamonds, blue squares, and black circles) in each panel indicate different repeated transect flight patterns for each wildfire smoke (Warneke et al., 2023).*

**R2-26: -  Line 364: "conclude" what?**

AC2-26: It refers to "*contributions of O to the TB formation*" and we made that explicit:

*(section 3.2) The increase in O with increasing smoke age was not explicitly observed in the current study, and further observations are needed to confirm contributions of O to the TB formation.*

**R2-27: -  Line 387: As mentioned before, what is implied by "in cloud"? Are these interstitial or residual TBs?**

AC2-27:  They are interstitial particles and we clarified that in the revised manuscript:

*(section 3.3) Some TBs and other organic particles collected in clouds (i.e., interstitial aerosol particles) in the Williams Flats TEM samples had thin layers or coatings (e.g., Figs. 2c, 2d, 11c, and 11d).*

**R2-28: -  Line 393: "trace of water" perhaps that should be "trace of water-soluble matter"? Same for line 397.**

AC2-28: Yes. We revised them as suggested:

*(section 3.3) Nonetheless, some TBs did not show any trace of water-soluble matter, indicating that they were relatively less hygroscopic than other particles (e.g., Fig. 11d).*

**R2-29: -  Line 398: why does the lack of change in shape indicate that the particles are not hygroscopic?**

AC2-29: We assumed that hygroscopic particles decrease viscosity in a high relative humidity condition and change their shape as shown in Cheng et al. (2021). We explained that in the revised text:

*(section 3.3) TBs with a trace of water-soluble matter retained their spherical shapes (Fig. 2d). These results suggest that although some TBs became CCN, they do not comprise highly hygroscopic substances and do not readily dissolve in water or change their shape at a high relative humidity condition (Cheng et al., 2021).*

**R2-30: -   Line 400: I am not convinced that this evidence is very strong as detailed here.**

AC2-30: We added some more discussion as described above (AC2-29) and tone down the conclusion:

*(section 3.3) In conclusion, our observation suggests that TBs can serve as CCNs but exhibit low efficiency.*

**R2-31: -   Line 415: "Increases of N" is this in TBs only?**

AC2-31: We found the N increase for both TBs and carbonaceous particles (Fig. 10). We clarified this in the revised manuscript.

*(section 3.4) Increases in N were observed with increasing age up to 6 h for both TBs and carbonaceous particles (Fig. 10), suggesting the formation of organic nitrogen compounds.*